# 3D-Printable Piezoelectric Composite Sensors for Acoustically Adapted Guided Ultrasonic Wave Detection [note 1]

**DOI:** 10.3390/s22186964

**Published:** 2022-09-14

**Authors:** Thomas Roloff, Rytis Mitkus, Jann Niklas Lion, Michael Sinapius

**Affiliations:** Institute of Mechanics and Adaptronics, Technische Universität Braunschweig, 38106 Braunschweig, Germany

**Keywords:** piezocomposite sensor, structural health monitoring, guided ultrasonic waves, sensor geometry, reflection analysis, acoustic impedance matching

## Abstract

Commercially available photopolymer resins can be combined with lead zirconate titanate (PZT) micrometer size piezoelectric particles to form 3D-printable suspensions that solidify under UV light. This in turn makes it possible to realize various non-standard sensor geometries which might bring benefits, such as increased piezoelectric output in specific conditions and less interference with incoming waves due to better acoustical adaptation compared to solid PZT ceramics. However, it is unclear whether piezoelectric composite materials are suitable for guided ultrasonic wave (GUW) detection, which is crucial for structural health monitoring (SHM) in different applications. In this study, thin piezoelectric composite sensors are tape casted, solidified under UV light, covered with electrodes, polarized in a high electric field and adhesively bonded onto an isotropic aluminum waveguide. This approach helps to demonstrate the capabilities of tape casting’s freedom to manufacture geometrically differently shaped, thin piezoelectric composite sensors for GUW detection. In an experimental study, thin two-dimensional piezoelectric composite sensors demonstrate successful detection of GUW for frequency-thickness products of up to 0.5 MHz mm. An analytical calculation of the maximum and minimum amplitudes for the ratio of the wavelength and the sensor length in wave propagation direction shows good agreement with the sensor-recorded signals. The output of the piezoelectric composite sensors and occurring reflections as measure for wave interactions are compared to commercial piezoelectric discs to evaluate their performance.

## 1. Introduction

In the emerging field of structural health monitoring (SHM) for large plate-like and complex thin-wall structures, guided ultrasonic waves (GUW) are state-of-the-research to detect damage and evaluate the condition of the structure [1,2,3,4]. GUW interfere with structural changes, e.g., stringers which leads to a complex wave field. To guarantee reliable measurements, direction sensitive actuation and sensing is under investigation [5,6,7,8]. The direction sensitivity is closely connected to the sensor size and dimensions ([9] p. 359ff.) and this results in the idea that the shape of the sensor also has an influence on the GUW detection.

Manufacturing methods such as 3D printing or tape casting allow almost free-form design of piezoelectric composite sensors that are solidified with UV light from suspensions made of lead zirconate titanate (PZT) particles dispersed in a photopolymer resin. Application-specific free-form designed, variable, direction and mode sensitive sensors could lead to a major extension of existing SHM setups. Additionally, they might possess better acoustic impedance matched with the waveguide, resulting in less interaction between applied sensors and incident GUW.

GUW are dispersive waves that appear in structures with two parallel free surfaces. They occur in symmetric and asymmetric modes and show displacements inside and on the surface of a structure. The particles perform in-plane and out-of-plane movements ([1] p. 198ff.), [10]. GUW are well suited for SHM applications due to their low damping over long distances ([4] p. 6).

The state of the art for GUW detection are solid piezoceramic discs ([1] p. 239ff.), but other piezoelectric materials exist, e.g., piezoelectric polymers or piezocomposite materials. Pure piezoceramics are stiff and brittle, cannot be applied on curved surfaces and often cause reflections of GUW due to their high acoustic impedance [11]. Piezoelectric polymers, like polyvinylidene fluoride (PVDF), are very flexible but offer a low electromechanic coupling and sensitivity. The aim of piezoelectric composite materials is to combine the advantages of both.

Manufacturing of piezoelectric composites in the field of SHM applications was introduced with the in-situ fabrication of piezoelectric wafer active sensors (PWAS) using a piezoelectric composite approach [12]. However, additive manufacturing methods of flexible piezoelectric composites are rarely mentioned in the SHM field. In recent research, inkjet printing of poly(vinylidene fluoride-trifluoroethylene) [13] and graphene/poly(amic acid)-based nanocomposite ink [14] is evaluated for ultrasonic imaging. Investigations on 3D printed piezoelectric composites are undertaken, but mostly in other subject fields, e.g., energy harvesting and ultrasonic or biomedical imaging [15,16]. In most cases, the piezoelectric material PZT is used as the active phase because of the very high piezoelectric properties compared to most other piezoelectric materials ($d_{33,PZT}=$ 225 - 590 Pc N^−1^) [17]. In particular, polymer- [18,19] and cement-based matrices [20] are used as the inactive phase of the composite.

While piezocomposite materials were addressed, the effect of the sensor geometry was not investigated, but modifications are possible just as mode-selective and directive actuators and sensors, e.g., sensor setups with interdigital electrodes [7] or multi-element setups acting as phased arrays [5,6,8]. The mode selectivity and directivity is strongly connected to the sensor geometry ([9] p. 359ff.).

The sensitivity of the sensors is of high importance for the performance of SHM applications. However, the sensors should only minimally interfere with the GUW to be detected as these interactions equally occur in other sensor’s signals which might mistakenly be interpreted just as damage. Damage results in a change in thickness or acoustic impedance, both leading to reflections, scattering, mode conversion, and amplitude attenuation [4]. Concerning the change in thickness, thin sensors are therefore favorable, which is contradictory to the sensitivity where the voltage generated increases with the sensor’s thickness [21]. However, the acoustic impedance can be adapted by changing the material properties, an advantage of piezocomposites where additives and different components can be used.

Reflections are minimised if the acoustic impedances of two adjacent media are equal [9,22,23]. The acoustic impedance depends on the material parameters density, Young’s modulus and Poisson’s ratio that can be influenced up to a certain extent by modifying the components of the piezocomposite material used in this study. Regarding the piezocomposite under investigation, it is assumed that it’s acoustic impedance is better acoustically matched to the aluminum waveguide than a solid PZT sensor.

This study experimentally investigates the applicability of tape casted piezoelectric composite sensors for GUW detection. In addition, the capabilities of free-form design using tape casting are evaluated, giving a first insight on the geometry dependency of the signal generation under GUW excitation. Furthermore, reflections at piezocomposite and PZT sensors are qualitatively compared to check for the acoustic impedance matching and minimal influences of the sensors on the wave propagation. GUW detection in an isotropic medium up to a frequency-thickness product of at least f·d= 0.5 MHz mm is validated, which makes these sensors applicable for SHM setups. The results give the appearance that the geometry of the sensor and sensor orientation with respect to the wave propagation direction play a key role in the sensor behavior. This behavior will be subject to future research. Additionally, the piezocomposite sensors result in less reflections when interacting with GUW than a solid PZT sensor which makes them better acoustically matched.

## 2. Materials and Methods

This section presents the theoretical background of GUW detection and reflection as well as the materials and processes used for the manufacturing of the piezocomposite sensors under investigation. The selection of different sensor geometries is explained prior to the description of the experimental setups to evaluate the detectability of GUW using piezocomposite sensors and for investigating occurring reflections when incoming GUW interfere with the surface-applied sensors to evaluate the acoustic impedance matching of piezocomposite and PZT sensors.

### 2.1. Theoretical Background on GUW Detection and Wave Interaction at Piezocomposite Sensors

When idealized as a plate capacitor, the voltage generated by a rectangular piezoelectric sensor under mechanical deformation can be calculated as follows:(1)$$U=\frac{d_{31}t_{s}Y_{s}}{\varepsilon_{33}^{\sigma }2a2b\left(1-\nu \right)}\underset{A}{\;\int \int \;}\left(\varepsilon_{x}+\varepsilon_{y}\right)\;dx\;dy$$
where $d_{31}$ denotes the piezoelectric charge coefficient, $t_{s}$ the sensor thickness, 2a the sensor length, 2b the sensor width, $Y_{s}$ the Young’s modulus of the sensor, $\varepsilon_{33}^{\sigma }$ the dielectric constant at constant mechanical stress, ν the Poisson’s ratio, and $\varepsilon_{x}$ and $\varepsilon_{y}$ the strains on the surface of the structure [21].

In the following consideration, a planar, one-dimensional wave field is assumed, generating strain in *x*-direction on the plate surface. All parameters except the sensor length are kept constant and the sensor is assumed to be a one-dimensional piezoelectric resonator. Then, the first amplitude maximum and minimum for the different modes occur at the following wavelengths λ, with a detailed description in ([1] p. 249ff.) and ([7] p. 21f.): (2)$$\begin{array}{cc} \mathrm{First}\;\mathrm{sensor}\;\mathrm{amplitude}\;\mathrm{maximum}\;\mathrm{at}:\;\lambda & =4a, \end{array}$$(3)$$\begin{array}{cc} \mathrm{First}\;\mathrm{sensor}\;\mathrm{amplitude}\;\mathrm{minimum}\;\mathrm{at}:\;\lambda & =2a. \end{array}$$

The previous statements show that the sensor performance, i.e., maximum voltage generated, depends on multiple parameters with size and geometry playing a key role.

The second parameter investigated in this study is the wave interaction between incident GUW and the piezocomposite sensors. This phenomenon is mainly influenced by different acoustic impedances. Assuming a far field condition for an acoustic source in a solid, i.e., the distance to the acoustic source is larger than 1.5 times the wavelength [24], the acoustic impedance *Z* is defined as the product of the density ρ and the phase velocity $c_{p}$
(4)$$Z=\rho c_{p}.$$

The acoustic impedance contributes to the reflection coefficient *R* that can be expressed in terms of acoustic impedances of two adjacent media $Z_{i}$ and $Z_{i+1}$ for a wave propagation perpendicular to the interface as follows [9,22,23]
(5)$$R=\frac{Z_{i+1}-Z_{i}}{Z_{i+1}+Z_{i}}.$$

Reflections are minimized if the acoustic impedances of the two adjacent media are equal. Therefore, the amplitude of reflected wave packages will be used in this study to evaluate the acoustic impedance matching between sensor and waveguide.

### 2.2. Sensor Manufacturing

The suspension used to manufacture sensors throughout this study consists of 20 vol% PZT particles (Material: PIC225, average particle size 1.6 μm, PI Ceramic, Lederhose, Germany) dispersed randomly in a photopolymer resin (High Temperature resin V2, Formlabs, Somerville, MA, USA) with a centrifugal vacuum mixer (Speedmixer DAC 700.2 VAC-P, Hauschild BmbH & Co. KG, Hamm, Germany). Materials are selected based on our previous studies [25,26]. No solvents or any other additives are used in suspension preparation. To achieve proper dispersion of the particles, the suspension is mixed under vacuum (20 mbar) in the centrifugal mixer for three times with the following parameters: 1 min at 900 min^−1^, 0.5 min at 1250 min^−1^ and 4 min at 1750 min^−1^. The dispersion quality is proven with scanning electron microscopy (SEM) imaging using a Helios G4 CX DualBeam system (FEI Company, Hillsboro, OR, USA). Figure 1 shows an SEM image of the piezocomposite at a magnification factor of 10,000× [27]. The cured photopolymer appears darker and the embedded PZT particles are brighter. No agglomerations occur and the particle distribution can be assumed as homogeneous. Because of the high density of PZT particles compared to the photopolymer ($\rho_{PZT}=$ 7.85 g cm^−3^, $\rho_{photopolymer}=$ 1.14 g cm^−3^), the suspension sediments in 24 h. Therefore, the suspension is remixed each time before sensor manufacturing.

The sensors are manufactured by tape casting. A PVC foil sticker (thickness 150 μm, Oraguard 270 G, ORAFOL Europe GmbH, Oranienburg, Germany) with the required sensor geometry pre-cut by a plotter is glued on glass. The suspension is filled on the sticker and tape casted manually with a metal blade held at 30° from vertical position. The glass with tape casted sensors is placed 50 mm below a UV light source (EQ CL30 LED Flood 405, Henkel AG & Co. KGaA, Düsseldorf, Germany) for 60 s for solidification.

Five individual micrometer measurements along the surface of the sensor are used to determine the respective sensor thicknesses required for dielectric measurements and polarization. Another pre-cut PVC sticker with the electrode geometry, i.e., 1 mm offset from outer edges of the sensor, is adhered onto the sensor. Silver coated copper (843AR Super Shield Silver Coated Copper Conductive Coating, MG Chemicals, Burlington, ON, Canada) is sprayed manually in two thin layers as an electrode. After drying, the sticker is peeled off, leaving the electrode on the sensor and the same procedure is repeated on the other side.

To polarize the sensors, a 55 kV mm^−1^ DC electric field is applied with a high voltage amplifier (gain 2000 VV^−1^, TREC 20/20C, Advanced Energy Industries, Inc., Denver, CO, USA) for 21 min in total (4 min ramp up, 16 min hold, 1 min ramp down) in a warm silicone oil at 65 °C. The poling voltage was controlled with the signal generator. After polarization, the sensors are dried with a paper towel and are left for a minimum of 24 h to dry further. Conductive silver ink (Silber-Leitlack, Busch GmbH & Co. KG, Viernheim, Germany) is used on the corner of each sensor to generate a single side access to both electrodes and ensure full and even sensor adhesion to the aluminum waveguide.

Table 1 presents the mechanical, dielectric and piezoelectric properties of both piezocomposite and commercial piezoelectric sensor (PRYY-1126, material PIC255, diameter 16 mm, ceramic height 200 μm), PI Ceramic GmbH, Lederhose, Germany) material. Their calculation and measurements are presented in detail in our previous publication [26]. The electromechanical coupling factor $k_{31}$ is calculated as follows
(6)$$k_{31}=\sqrt{\frac{d_{31}^{2}}{s_{11}^{E}\varepsilon_{33}^{T}}},$$
where $s_{11}^{E}$ denotes the elastic compliance at no electric field (inverse of Young’s modulus) and $\varepsilon_{33}^{T}$ the dielectric constant at no mechanical stress.

### 2.3. Sensor Geometry Selection

For comparability, all sensor electrode areas are set to 324 mm^2^. The overall size of the sensors with different geometries may vary due to the 1 mm offset. The mean sensor thickness is 129.9 μm and the average thickness of both applied electrodes is 44.3 μm. In addition to conventional geometries (square and circle), the more complex geometry of an annulus segment is investigated. Its radii are adapted to the expected propagating wavefront of a circular actuator. Figure 2 shows the respective sensor geometries and a commercial circular piezoceramic sensor in respective orientation to the wave propagation direction [27].

### 2.4. Experimental Setup for the Determination of Detectable GUW Signals

The test setup is shown in Figure 3 [27]. A square aluminum plate (material 3.3535, density 2.66 g m^−3^, Young’s modulus 70.5 GPa, Poisson’s ratio 0.33) with an area of 1 m × 1 m and a thickness of 2 mm is used as a waveguide. A piezoceramic disc transducer PRYY-1126 is used for excitation and adhesively bonded to the center of the plate with cyanoacrylate. Due to the circular ceramic, the wave field is assumed to have a concentrically propagating circular wave front. The sensors are equally glued to the aluminum plate in a circular arrangement with the sensor’s geometric center on a circle with a radius of 156 mm around the actuator. The sensors under investigation will be placed in two orientations with respect to the wavefront except for the circular ones, see Figure 2. A PicoScope 5442B is used in combination with a laptop to serve as a signal generator to provide the excitation signal and the amplification is realized using a high voltage amplifier (WMA-300, Falco Systems, Katwijk aan Zee, The Netherlands). The laptop with the PicoScope also acquires the measurement data.

For excitation, a 5-cycle, Hanning-windowed sine burst is used. The investigated burst center frequencies range from 5 kHz to 200 kHz with an interval of 5 kHz and from 200 kHz to 250 kHz with an interval of 25 kHz. Due to the short distance between the actuator and the sensors, no temporal separation of the $S_{0}$ and $A_{0}$ modes is possible. Therefore, the peak-to-peak voltage amplitude $U_{pp}$ is measured in a time window from the calculated start of the faster $S_{0}$ to the end of the slower $A_{0}$ mode, see Figure 4 [27]. To generate comparable sensor signals, a normalization is performed. The signals are normalized using the sensor’s thicknesses, a factor to compensate the capacity loss due to polarization errors and a factor to compensate for the amplifier behavior, as the amplification factor decreases with increasing frequency depending on the capacitive load of the actuator. The sensor’s thickness is evaluated as mentioned above. The capacity loss compensation is determined by comparing the piezocomposite’s capacity prior to and after the polarization process to compensate possible damage of the electrodes due to the high voltages during polarization. The amplifier behavior is considered by measuring the excitation signal after amplification using a voltage probe and normalize it to the maximum excitation voltage at the lowest frequency under investigation.

### 2.5. Experimental Setup for the Determination of GUW Reflections at Applied Sensors

This section addresses the experimental setup to qualitatively compare different surface-applied sensors with regard to their acoustic impedance matching with the aluminum waveguide. A three-dimensional scanning laser Doppler vibrometer (SLDV) technique is used to detect reflections at the sensors, an indicator for a change in acoustic impedance.

A sketch of the experimental setup is depicted in Figure 5. The same aluminum plate as described in Section 2.4 is used as waveguide. In this case, retroreflective tape is applied in lines on the path from the actuator to behind four sensors under investigation: PZT sensor, cf. Figure 2b, square sensor orientation 1, cf. Figure 2c, square sensor orientation 2, cf. Figure 2d, and circular sensor, cf. Figure 2e. These sensors are selected to compare the 3D-printable piezocomposite sensors to a commercial material and to investigate the influence of the sensor’s geometry. The retroreflective tape increases the reflectivity for a better signal-to-noise ratio in the SLDV signal. Again, a computer and a signal generator are used to generate the excitation signal described in Section 2.4 while the same high-voltage amplifier amplifies the signal for actuation. This time, a burst center frequency of 300 kHz is used to excite GUW with a wavelength in the order of magnitude of the sensor’s dimensions as this supports detectable interactions, cf. determined wavelengths in Table 2.

Three individual SLDV measurements along the depicted measuring path are performed consecutively with angles of approx. 90° between all measurements. A subsequent coordinate transformation according to [28] is performed to separate the in-plane and out-of-plane components to investigate the $A_{0}$ and $S_{0}$ mode separately. The data is bandpass filtered with cut-off frequencies of 200 kHz and 400 kHz and evaluated in form of B-Scans by plotting the temporal data for every measurement point in a two-dimensional plot using a color scale to show the amplitude in the third dimension.

The method used to quantify occurring reflections is adapted from [29,30] and presented in Section 3.2.

## 3. Results

The overall aim of this study is to prove in general that tape casted piezocomposite sensors are capable of detecting GUW in isotropic media. In addition, their influence on the propagating wave field is investigated by comparing occurring reflections at applied piezocomposite sensors to a commercial PZT transducer.

### 3.1. GUW Detection Using Tape Casted Piezocomposite Sensors

Figure 4 shows an exemplary time signal of the excitation signal and the generated voltage *U* [V] by a piezocomposite sensor in comparison with a commercial piezoelectric disc sensor. This data was generated using the setup described in Section 2.4. Although the generated voltage by the piezocomposite sensor is two powers of ten lower, it shows the same qualitative behavior as the commercial one. This leads to the conclusion that the presented tape casted, but also 3D-printable piezocomposite sensors are capable of detecting GUW and are suitable to be used in SHM setups. The higher amplitude of the piezoceramic sensor is due to a higher thickness ($h_{PRYY-1126}=$ 200 μm, $h_{piezocomposite}=$ 129.9 μm), stiffness, and piezoelectric charge coefficient, cf. Equation (Equation 1) and Table 1. The piezoelectric charge coefficient $d_{31}$ of the commercial PRYY-1126 sensor is approx. 200 times higher than the one of the piezocomposite sensor.

Apart from the general proof that the piezocomposite sensors are capable of detecting GUW, a first insight on the frequency and geometry dependency of the generated voltage is presented in the following paragraphs. The results for the sensors shown in Figure 2, manufactured and measured as described in Section 2, are presented in Figure 6 by plotting the normalized maximum voltage generated by the respective sensors over the frequency–thickness product referring to the burst center frequency of the excitation signal [27].

All sensors show a qualitatively similar behavior with maxima and minima at different frequencies, as expected in Section 1. According to Equations (Equation 2) and (Equation 3) and the assumption of a one-dimensional wave propagation, the expected frequencies/wavelengths for a maximum or minimum amplitude for a given sensor length are calculated and shown as solid and dotted vertical lines, respectively. Figure 2 shows the assumed effective sensor lengths in wave propagation direction. The frequency dependent wavelengths of the aluminum waveguide are calculated with the material properties stated in Section 2.4. The analytical results for the expected maximum and minimum amplitudes fit well with the measurements of the annulus segment shaped sensor and the square shape one in orientation 1. The two circular sensors show slight deviations from the calculated extremes and the results of the rotated square sensor (orientation 2) deviate most from the analytical calculations. Possible explanations are erroneous material properties in the analytical solution, a superposition of the $A_{0}$ and $S_{0}$ mode as the group velocities do not differ enough for wave package separation and most likely a wrong estimation of the effective sensor length and an erroneous assumption of a one-dimensional wave field.

Although the qualitative behavior is the same, the amplitudes differ between the different sensors. The commercially available piezoceramic sensor shows higher amplitudes than the custom piezocomposite sensors over the whole investigated frequency range. This phenomenon is already explained above. Furthermore, the maxima of the annulus segment shaped sensor are higher than for the standard geometries, i.e., circular and square shape. This might give the impression that a short effective sensor length leads to higher amplitudes. This is only valid for a constant sensor width, cf. Equation (Equation 1), and furthermore, the square shaped sensor shows higher amplitudes for orientation 2 with a higher effective sensor length than for orientation 1. Moreover, the two measurements for the square sensor in orientation 2 differ considerably from one another. This shows that more profound investigations are necessary to reliably characterize these sensors.

Another interesting finding is that the annulus segment shaped sensor shows better performance in orientation 2 although this is the one that is not adapted to the expected wave field.

Two possible explanations for the slight differences in the amplitudes between all of the piezocomposite sensors are to be discussed in further research. First, the adapted radii of the annulus segment shaped sensor to the expected wave field might lead to a higher instantaneous reflection of the incoming wave. This would result in a reduced vibration amplitude inside the sensor. Second, the emitted wave field from the actuator might not be concentric. It is a ceramic with a wrap-around contact that allows single side access to the electrodes but results in an non-circular electrode surface on the top [31]. Moreover, the resonance behavior of a piezoelectric actuator disc results in an inhomogeneous wave field emission as well [32]. If the wave field is not concentric, the sensors undergo different excitations in their circular setup.

Generally speaking, the piezocomposite sensors are capable of detecting GUW in isotropic media for frequency-thickness products of up to 0.5 MHz mm. However, their sensitivity is more than an order of magnitude smaller than that of a solid PZT ceramic, and the geometry and geometric orientation with regard to the wave propagation direction play a role, too. Further improvement of the material combination is necessary to increase the piezoelectric charge coefficient which could eventually enable the piezocomposites to be equally used as actuators. So far, GUW cannot be emitted with a detectable amplitude by the piezocomposites under investigation.

### 3.2. GUW Reflections at Applied Piezocomposite Sensors

Surface-applied PZT and differently shaped piezocomposite sensors are qualitatively compared with regard to their acoustic impedance matching. In the following, the method to determine reflections of GUW when interacting with the sensors and it’s results are shown as an example for the PZT sensor due to best visibility. The same method is applied to the square sensor in orientation 1 and orientation 2 as well as the circular piezocomposite sensor. The results are compared at the end of this section.

The B-scans in Figure 7a,b represent the measured in-plane and out-of-plane component of the wave propagation along the measuring path. The data is generated by coordinate transformation as stated in Section 2.5. The B-scans represent the wave propagation using a two-dimensional spatial-temporal presentation with a color scale as third dimension to display the structure’s velocity in [m s^−1^]. The visualized wave propagation contains all incident and reflected wave packages.

It is visible that the $S_{0}$ mode has a higher in-plane component while the $A_{0}$ mode has a higher out-of-plane component. Following this, investigations on the $S_{0}$ mode will be performed using the in-plane component, while the $A_{0}$ mode will be investigated using the out-of-plane component.

B-scans can be used to measure the group velocity of wave packages by tracking the wave packages maximum, i.e., the slope of the wave package propagation in the B-scan. The determined values are listed in Table 2 and fit well with the theoretically determined values that are generated as stated in Section 3.1. These group velocity can be used to indicate the theoretical course of the incident $A_{0}$ and $S_{0}$ mode wave packages in Figure 7a,b with solid red and yellow lines, respectively.

In addition, the position of the PZT sensor is marked and its spatial start and end act as source of reflections as the applied sensor leads to a change in thickness and material properties. The theoretically occurring course of reflected $A_{0}$ and $S_{0}$ mode wave packages are indicated in dashed red and yellow lines, respectively. Additionally, $x_{investigate}$ is shown for reflection quantification described later in this section.

Figure 7c,d show quadrants 1 and 4 of the spatial-temporal two-dimensional Fourier transformation of the B-scans in Figure 7a,b. The data is represented in the frequency-wave number-plane with a color scale to indicate the amplitudes. The positive frequency range describes the physically interpretable data and the result of the bandpass filtering is clearly visible in the limited frequency bandwidth with significant amplitude. Positive wave numbers indicate incident wave propagation while negative wave numbers indicate reflected wave packages.

Four areas with elevations are visible. Two higher ones with positive and two lower ones with negative wave numbers. It is clearly visible that the incident waves have higher amplitudes than the reflected waves. Again, the $S_{0}$ mode has a higher in-plane component than the $A_{0}$ mode, cf. Figure 7a, and vice versa for the out-of-plane component, cf. Figure 7b.

Dashed white boxes indicate identified regions of incident and reflected $S_{0}$ and $A_{0}$ modes as entitled in the Figures. The identified wave numbers at $f_{c}=$ 300 kHz in the middle of the identified elevations in the dashed white boxes correspond to the occurring wavelengths. The experimentally determined wavelengths are listed in Table 2 and are in good accordance with the theoretically determined ones as stated in Section 3.1. As expected, the wavelengths are in the order of magnitude of the sensor’s dimensions and therefore, reflections should occur. The dashed white lines additionally serve as boundaries used for applying two-dimensional Hanning windows for each detected incident and reflected wave mode, respectively. This wave number-frequency filtering is done prior to performing an inverse Fourier transformation on this filtered data to separate incident and reflected $S_{0}$ and $A_{0}$ modes in the spatial-temporal domain.

Figure 8 shows the B-scans after performing the inverse Fourier transformation on the two-dimensionally Hanning windowed filtered data from Figure 7c,d.

It can be seen that the separated incident $S_{0}$ and $A_{0}$ wave packages follow the theoretically determined wave propagation, cf. solid yellow line in Figure 8a for the $S_{0}$ mode and solid red line in Figure 8b for the $A_{0}$ mode. Additionally, the reflected wave packages follow the theoretically determined wave propagation as well, cf. wave packages in Figure 8c,d with the corresponding dashed lines.

However, the reflections occur at the beginning and end of the PZT sensor in the measuring path as at both locations, thickness and material properties change. Furthermore, the incoming $S_{0}$ mode is reflected as both an $S_{0}$ and an $A_{0}$ wave package at the obstacle. This phenomenon has also been observed at surface-applied stringers on structures which also locally increase the structure’s thickness [4].

So far, the method follows the work presented in [29,30]. The adaptation in this work is done by evaluating selected temporal data from the filtered B-scans. For qualitative comparison of reflections at different sensors, the filtered temporal data will be extracted for the three different sensor shapes and investigated at $x_{investigate}$, a position in the measuring path, 50 mm before the sensors under investigation, cf. Table 3.

Figure 9 shows exemplary filtered temporal data at $x_{investigate}$ for the PZT sensor to demonstrate the determination of the reflection’s amplitude, cf. Figure 8. It is important to note that in this work only the following reflections will be compared: A reflected $S_{0}$ mode from an incident $S_{0}$ mode and a reflected $A_{0}$ mode from incident $A_{0}$ mode. Comparisons including mode conversion will be addressed in future work.

The presented procedure will be repeated for all four sensors under investigation and the results are compared qualitatively.

Figure 9a shows the filtered incident and reflected $S_{0}$ mode at $x_{investigate}$ from the in-plane component of the wave field, cf. Figure 8a,c. The two *y*-axes show that the reflection is one order of magnitude smaller than the incident wave package. Using the group velocities determined above and listed in Table 2 the source of the reflected wave packages can be located. The time shift between the maxima in combination with the group velocity result in a distance approximation. As the distance to the start of the sensor is known from the setup, the wave package corresponding to a reflection at the respective sensor can be identified.

For the in-plane component at $x_{investigate}=$ 98 mm, the amplitude of the reflected $S_{0}$ wave package at the PZT sensor is 4.60% of the incident $S_{0}$ wave package’s amplitude, cf. Figure 9a. The same procedure is applicable for the $A_{0}$ mode and out-of-plane component in Figure 9b. Here, the ratio between reflected and incident $A_{0}$ wave package is 1.76%. In addition, the mode conversion described above can equally be detected. The incoming $S_{0}$ mode results in a reflected $A_{0}$ mode as well. This reflection arrives earlier than the $A_{0}$–$A_{0}$ reflection as the $S_{0}$ group velocity is higher so that the time of reflection at the sensor is earlier. However, these mode conversions will be looked at in future research.

Table 3 shows the results for the reflection analysis at different surface-applied sensors when interacting with GUW of a burst center frequency of $f_{c}=$ 300 kHz. The ratio between the magnitude of the incident and reflected wave package is listed for a spatial point 50 mm in front of the respective obstacles. It is important to note that these values can only be interpreted and compared qualitatively as they are only valid for one position ($x_{investigate}$). Three conclusions result.

First, the $S_{0}$–$S_{0}$ reflection in the in-plane component is higher than the $A_{0}$–$A_{0}$ reflection in the out-of-plane component. This leads to the conclusion that the $S_{0}$ mode is more sensitive to changes in the structure’s thickness as all configurations have in common that an applied sensor locally increases the waveguide’s thickness. Regarding the determined wavelengths, a higher influence on the $A_{0}$ mode was expected as it’s wavelength is three times smaller than the sensor dimensions while the one of the $S_{0}$ mode is nearly equal to the sensor dimensions.

Second, reflections at the PZT sensor are higher than for all piezocomposite sensors under investigation. The PZT is slightly thicker but the square sensors are larger compared to the GUW’s wavelength. As the main changing parameter is the material, it can be concluded that the piezocomposite is acoustically better adapted to the aluminum waveguide than PZT. Therefore, it results in less reflections when interacting with incoming GUW.

Third, the reflections at the square sensor with orientation 1, cf. Figure 2c, are higher than for orientation 2, cf. Figure 2d. The former has a sensor edge perpendicular to the wave propagation direction while the latter is pointed with it’s tip towards the incoming wave. The perpendicular edge logically leads to higher reflections. It can be concluded that a reasonable orientation of a sensor in wave propagation direction might reduce reflections. However, this study ignores scattering which might be higher for the square sensor in orientation 2 and scattering can equally be harmful for GUW detection in sensor networks for SHM applications.

In general, the piezocomposite is acoustically better adapted to the waveguide compared to PZT and therefore results in less reflections when interacting with GUW. Hence, in this category it should be favoured. The geometry plays a role as well but might have negative influence if reflections in one direction are reduced, but scattering is increased which equally results in parasitic signals in a sensor network.

## 4. Conclusions

In this study, the detection of GUW in isotropic waveguides using tape casted piezoceramic composite sensors based on photopolymers is validated. This is experimentally shown for an isotropic aluminum for a frequency-thickness product of up to at least 0.5 MHz mm. The experimental results revealed that different piezocomposite sensor sizes and shapes show different sensitivities. Concerning the size, the ratio between effective sensor length and wavelength plays a key role with maximum signal amplitude when multiples of the half wavelength fit into the sensor in wave propagation direction. Additionally, it was shown that different orientations of sensors with the same geometry can lead to higher signals generated. However, their sensitivity can not reach the one of solid PZT discs yet, which are generating voltages 50× higher. Nevertheless, the geometry dependency is promising for future design of optimized piezocomposite GUW sensors.

A reflection analysis revealed less reflections of an incoming GUW package when interacting with piezocomposite sensors compared to a solid PZT sensor. This indicates that they are better matched to the waveguide concerning the acoustic impedance. This is favorable for sensor design as reflections at sensors additionally occur in sensor network signals and might interfere with the signals from damage to be detected.

To answer the open questions concerning the geometry dependency and to reach new forms of sensors, the following research topics need to be addressed in future publications:Optimize the material properties to increase the piezoelectric sensitivity and further minimize reflections when interacting with GUW.Replace PZT by lead-free particles such as barium titanate or potassium sodium niobate.Consider geometry rather than only the estimated effective sensor length as a criterion for sensor performance. This can be done, providing an analytical model to link the sensor response to an excited GUW wave field, e.g., a 3D linear elasticity model as presented in [33].Realizing a defined characterization environment to extract the sensor behavior in a sort of frequency response function that is independent from the sensor’s excitation.Design a concept for variable, direction-sensitive, and mode-selective sensors.Address mode conversion in reflection analysis for further insight in wave propagation interference by applied sensors.

## Figures and Tables

**Figure 1 sensors-22-06964-f001:**
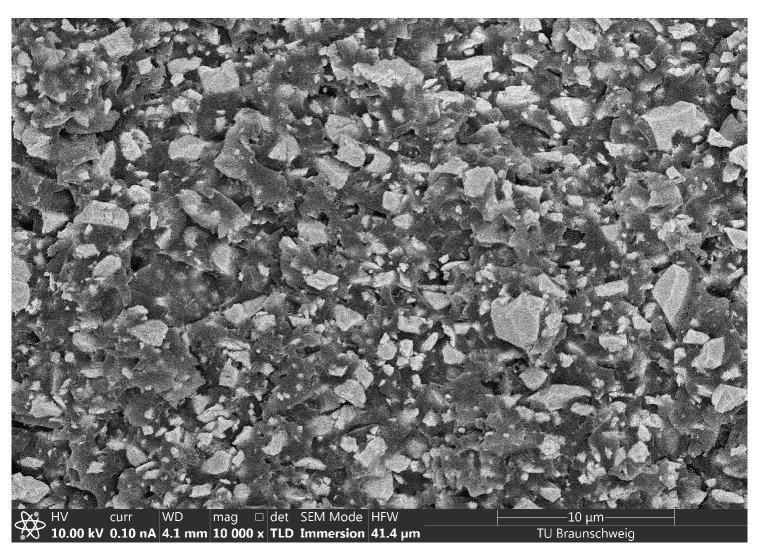
Scanning Electron Microscopy image of 20 vol% PZT/photopolymer composite shows homogeneous particle dispersion and no agglomerations.

**Figure 2 sensors-22-06964-f002:**
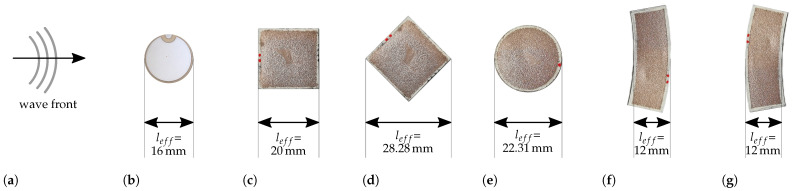
Sensor shapes under investigation with regard to the wave propagation direction and assumptions concerning the effective sensor length in wave propagation direction. (**a**) Wave front of a circular wave field. (**b**) Commercial Piezoceramic PRYY-1126. (**c**) Sensor shape: square, Orientation: 1. (**d**) Sensor shape: square, Orientation: 2. (**e**) Sensor shape: circle, Orientation: -. (**f**) Sensor shape: annulus segment, Orientation: 1. (**g**) Sensor shape: annulus segment, Orientation: 2.

**Figure 3 sensors-22-06964-f003:**
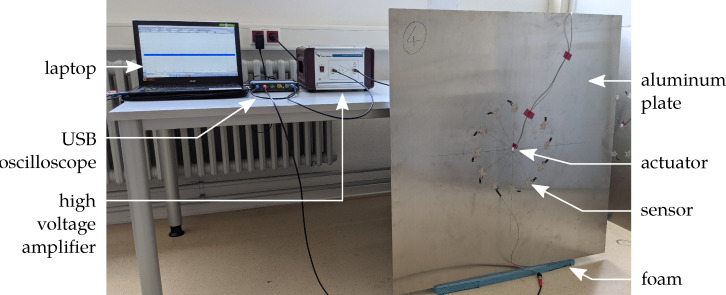
Test setup to determine the peak-to-peak voltage of the sensors under GUW excitation.

**Figure 4 sensors-22-06964-f004:**
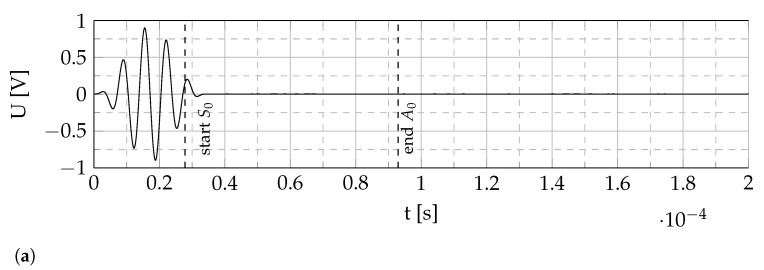
Exemplary signal comparison of a commercial piezoelectric disc sensor and a piezocomposite square sensor under GUW excitation. Dashed lines indicate the estimated start and end times of the faster $S_{0}$ and slower $A_{0}$ wave package, respectively, at the sensor location. (**a**) Excitation signal before amplification: 5-cycle hanning windowed sine burst ($f_{c}=$ 150 kHz). (**b**) Sensor signals.

**Figure 5 sensors-22-06964-f005:**
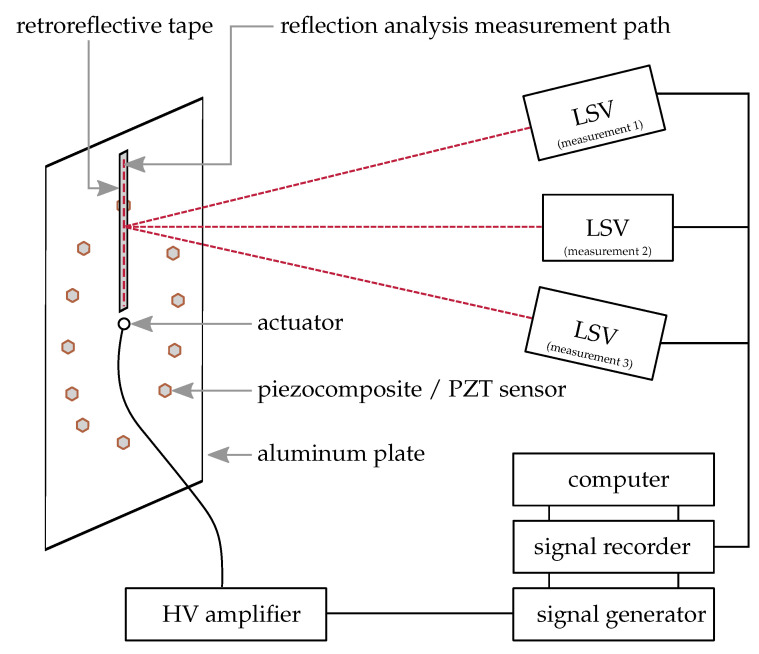
Experimental setup of the three-dimensional scanning laser Doppler vibrometry technique for separate measurement of the in-plane and out-of-plane components of the wave propagation to determine the GUW reflections at the applied sensors.

**Figure 6 sensors-22-06964-f006:**
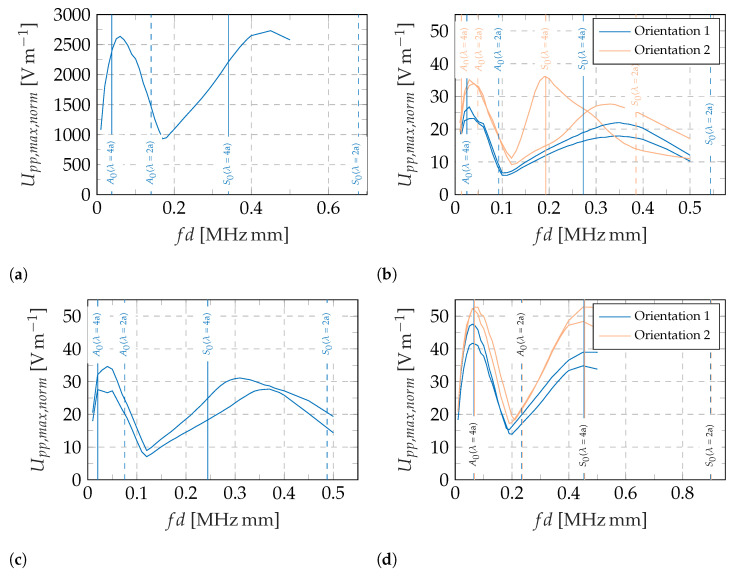
Experimentally determined peak-to-peak voltages for different sensor types, shapes, and orientations under GUW excitation in a 2 mm aluminum plate, analytically calculated amplitude maxima (solid vertical lines) and minima (dashed vertical lines) based on estimated effective sensor lengths in wave propagation direction, cf. Figure 2 and Equations (Equation 2) and (Equation 3). (**a**) Commercial solid piezoceramic sensor, Figure 2b. (**b**) Piezocomposite sensor: square, Figure 2c,d. (**c**) Piezocomposite sensor: circle, Figure 2e. (**d**) Piezocomposite sensor: annulus segment, Figure 2f,g.

**Figure 7 sensors-22-06964-f007:**
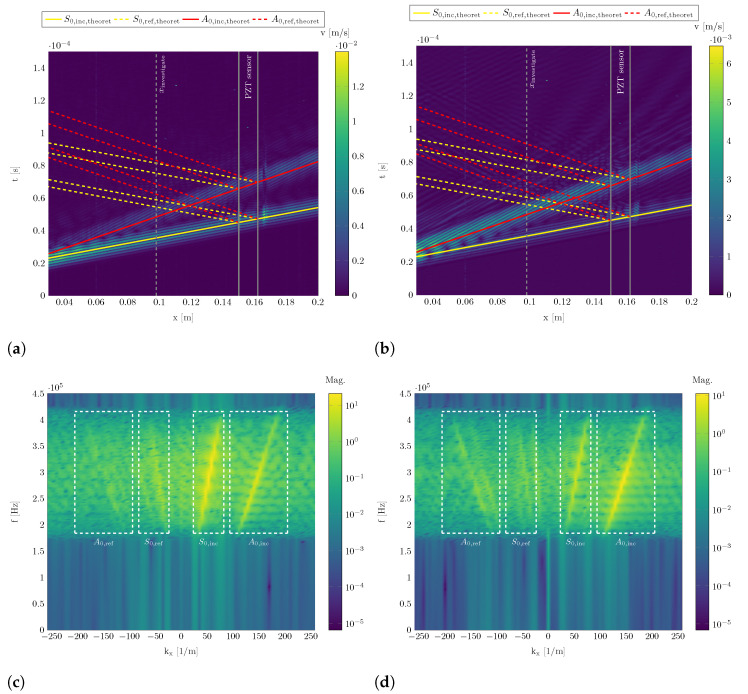
B-scans and two-dimensional spatial-temporal Fourier transformation of three-dimensional SLDV data at $f_{c}=$ 300 kHZ in the 2 mm aluminum plate along the measuring path containing the PZT sensor (Figure 2b). (**a**) B-Scan (In-plane component). (**b**) B-Scan (Out-of-plane component). (**c**) Two-dimensional FFT of in-plane data (Quadrants 1 and 4). (**d**) Two-dimensional FFT of out-of-plane data (Quadrants 1 and 4).

**Figure 8 sensors-22-06964-f008:**
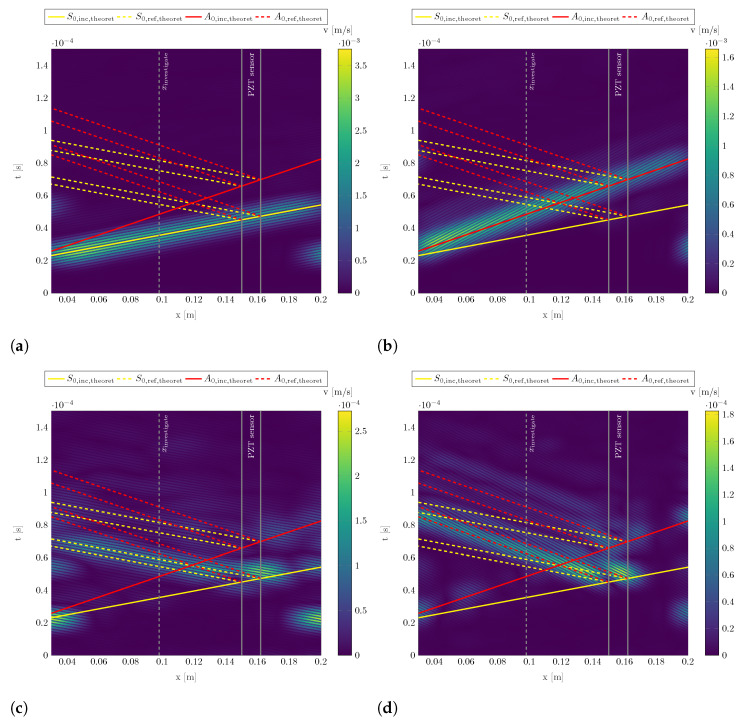
Filtered B-scans for identifying incident and reflected $S_{0}$ and $A_{0}$ modes for an excitation with a burst center frequency of $f_{c}=$300 kHz. 2mm aluminum plate with surface-applied PZT ceramic. (**a**) Filtered incident $S_{0}$ mode from in-plane data. (**b**) Filtered incident $A_{0}$ mode from out-of-plane data. (**c**) Filtered reflected $S_{0}$ mode from in-plane data. (**d**) Filtered reflected $A_{0}$ mode from out-of-plane data.

**Figure 9 sensors-22-06964-f009:**
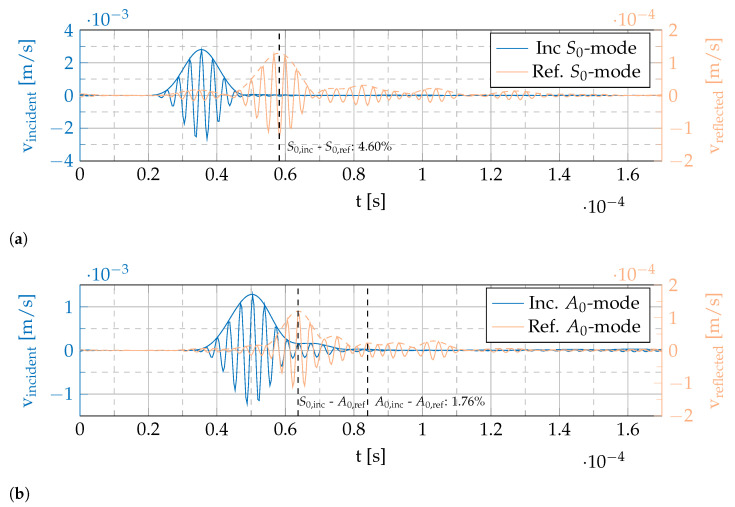
Exemplary experimentally determined reflection quantification at $x_{investigate}=$ 98 mm using filtered signals from three-dimensional SLDV measurement in 2 mm aluminum plate with surface-applied PZT sensor. (**a**) Filtered incident and reflected $S_{0}$ proportions of in-plane data. (**b**) Filtered incident and reflected $A_{0}$ proportions of out-of-plane data.

**Table 1 sensors-22-06964-t001:** Properties of the piezoelectric materials under investigation.

Material	Young’s Modulus (perp. to pol. dir.) *Y* [GPa]	Dielectric Permittivity at 1 kHz $\varepsilon_{33}$ [nF m${}^{-1}$]	Piezoelectric Charge Constant $d_{31}$ [pC N${}^{-1}$]	Electromechanical Coupling Factor $k_{31}$ [-]
Custom piezocomposite	1.8 ± 0.2	0.08 ± 0.002	−0.92 ± 0.13	0.0044 ± 0.00091
Commercial ceramic (PIC255) [11]	62.5	15.9	−180	0.35

**Table 2 sensors-22-06964-t002:** Experimentally and numerically determined group velocities and wavelengths in the 2 mm aluminum (material: 3.3535) waveguide at $f_{c}=$300 kHz.

Measured Variable	Parameter	Dimension	$S_{0}$-Mode (num.)	$S_{0}$-Mode (exp.)	$A_{0}$-Mode (num.)	$A_{0}$-Mode (exp.)
Group velocity	mean	[m s^−1^]	5368.71	5498.4	3022.04	2987.0
	standard dev.	[%]	-	2.10	-	1.06
Wavelength	mean	[mm]	0.0181	0.0182	0.0067	0.0067
	standard dev.	[%]	-	1.42	-	0.74

**Table 3 sensors-22-06964-t003:** Experimentally determined reflections at applied sensors at $f_{c}=$ 300 kHz at a distance of 5 cm to the respective sensors. Percentage value of ratio between incident and reflected wave package amplitudes.

Type of Reflection	Data Base	Dimension	PZT	Square (Orien. 1)	Square (Orien. 2)	Circle
$S_{0}$–$S_{0}$	in-plane data	[%]	4.60	2.09	1.58	0.95–2.50 (*)
$A_{0}$–$A_{0}$	out-of-plane data	[%]	1.76	1.17	1.22	1.75
$x_{investigate}$	-	[mm]	98	96	92	95

(*) Smeared reflection with varying amplitude.

## Data Availability

The data presented in this study are available on request from the corresponding author.

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
