# Peer review of "3D-Printable Piezoelectric Composite Sensors for Acoustically Adapted Guided Ultrasonic Wave Detectionâ€"

_sensors, 2022, doi:10.3390/s22186964_

Round 1
Reviewer 1 Report
The manuscript is well prepared and the reviewer recommends it for publication after minor revision.
In the conclusion, the authors should list the findings rather than simply saying that "different piezocomposite sensor sizes and shapes show different sensitivities and that their sensitivity can not reach the one of solid PZT discs yet".
Reviewer 2 Report
The presented work concerns research on the applicability of 3D printable piezoelectric composite sensors for guided ultrasonic wave detection GUW. Sensors of various shapes were made and tests were carried out in terms of their usefulness for GUW detection. I found no substantive errors in the paper. However, the authors should standardise the naming of the manufacturing method-3D printing or tape casting (both methods are different). I would suggest using the name 'tape casting'.
Reviewer 3 Report
In the reviewed manuscript, the authors are presenting novel findings in the field of additively manufactured ultrasonic sensors. It is shown that the developed piezocomposite sensors are not only capable to detect elastic guided waves but also allows for lower unnecessary reflections due to their better acoustic match with the host metallic structure. In general, the paper is perfectly fitting the scope of the Sensors journal and could be of high potential interest for the scholars within the ultrasonic-based NDT/SHM communities. Nevertheless, some revision might be suggested that could make the results more meaningful.
Particular suggestions and comments are as follows:
1) It is slightly confusing to see certain amount of formulas in the Introduction section. It might be suggested to gather all the preliminary theoretical notes into a separate section.
2) Since the developed piezocomposite devices are tested only as sensors, it might be suggested to mention in the Introduction some recent advancements in other additive-based techniques for ultrasonic sensors (i.e., ink printing https://doi.org/10.1063/5.0048444, https://doi.org/10.1016/j.coco.2021.100937).
3) Could these manufactured piezocomposite sensors perform as actuators? What would be the amplitude of the excited wave signals?
4) Certain clarification for the description of the experimental setup might be suggested:
- was this PIC225 material acquired directly from PI Ceramic as PZT particles or milled somehow by the authors?
- how the vacuum mentioned on Page 4 was achieved?
- what were the particular devices (manufacturer, type) used for SEM imaging, estimation of sensor thickness, their polarization?
5) In the last sentence of the Subsection 2.3, certain compensation factors are mentioned. How are they evaluated?
6) When evaluating reflections (i.e., Table 3) the results for square piezocomposite sensors are compared the data obtained for circular PZT-disc. For better consistence, the results for piezocomposite disc might be also added here.
7) In the last paragraph of Page 8, it is mentioned that "Second, the emitted wave field from the actuator might not be concentric.". Paper https://doi.org/10.1016/j.sna.2011.11.008 probably might be mentioned here.
Round 2
Reviewer 3 Report
The authors have carefully responded to all the comments. In its current form, the manuscript is a nice example of a solid and topical research with interesting and promising results. Thus, it should be accepted for publishing in Sensors. I could only wish the authors good luck in their further research activities.